# A Lymph Node Ratio Model for Prognosis of Patients with Pancreatic Neuroendocrine Tumors

**DOI:** 10.3390/biomedicines11020407

**Published:** 2023-01-30

**Authors:** Esther Osher, Eiman Shalabna, Joseph M. Klausner, Yona Greenman, Naftali Stern, Oren Shibolet, Erez Scapa, Oz Yakir, Dana Ben-Ami Shor, Iddo Bar-Yishay, Sivan Shamai, Yael Sofer, Nir Lubezky, Yaacov Goykhman, Guy Lahat, Ido Wolf, Sharon Pelles, Asaf Aizic, Arye Blachar, Ravit Geva

**Affiliations:** 1Institute of Endocrinology, Metabolism and Hypertension, Tel Aviv Medical Center, School of Medicine, Sackler Faculty of Medicine, Tel Aviv University, Tel Aviv 64239, Israel; 2Department of Oncology, Tel Aviv Medical Center, School of Medicine, Sackler Faculty of Medicine, Tel Aviv University, Tel Aviv 64239, Israel; 3Department of Surgery, Tel Aviv Medical Center, School of Medicine, Sackler Faculty of Medicine, Tel Aviv University, Tel Aviv 64239, Israel; 4Department of Gastroenterology, Tel Aviv Medical Center, School of Medicine, Sackler Faculty of Medicine, Tel Aviv University, Tel Aviv 64239, Israel; 5Institute of Pathology, Tel Aviv Medical Center, School of Medicine, Sackler Faculty of Medicine, Tel Aviv University, Tel Aviv 64239, Israel; 6Institute of Radiology, Tel Aviv Medical Center, School of Medicine, Sackler Faculty of Medicine, Tel Aviv University, Tel Aviv 64239, Israel

**Keywords:** lymph node ratio, pancreatic, neuroendocrine tumor, ki-67

## Abstract

The objective of this study was to determine the prognostic value of lymph node (LN) involvement and the LN ratio (LNR) and their effect on recurrence rates and survival in patients with pancreatic neuroendocrine tumors (PNETs) undergoing surgery. This single-center retrospective study reviewed the medical records of 95 consecutive patients diagnosed with PNETs who underwent surgery at our medical center between 1997 and 2017. The retrieved information included patient demographics, pathology reports, treatments, and oncological outcomes. Results: 95 consecutive potentially suitable patients were identified. The 78 patients with PNETs who underwent surgery and for whom there was adequate data were included in the analysis. Their mean ± standard deviation age at diagnosis was 57.4 ± 13.4 years (range 20–82), and there were 50 males (64%) and 28 females (36%). 23 patients (30%) had LN metastases (N1). The 2.5- and 5-year disease-free survival (DFS) rates for the entire cohort were 79.5% and 71.8%, respectively, and their 2- and 5-year overall survival (OS) rates were 85.9% and 82.1%, respectively. The optimal value of the LNR was 0.1603, which correlated with the outcome (2-year OS *p* = 0.002 HR = 13.4 and 5-year DFS *p* = 0.016 HR = 7.2, respectively, and 5-year OS and 5-year DFS *p* = 0.004 HR = 9 and *p* = 0.001 HR = 10.6, respectively). However, the multivariate analysis failed to show that the LNR was an independent prognostic factor in PNETs. Patients with PNETs grade and stage are known key prognostic factors influencing OS and DFS. According to our results, LNR failed to be an independent prognostic factor.

## 1. Introduction

Pancreatic neuroendocrine tumors (PNETs) are rare and account for ~3–5% of pancreatic tumors, with a yearly incidence of 5.25 per 100,000 people in the United States [1]. The natural history of PNETs is highly variable, extending from indolent behavior to an aggressive course and from local invasion to distant metastasis. Factors associated with a less favorable prognosis are non-functioning tumor, size, higher tumor grade and stage, advanced age, and distant metastases. Risk factors were taken from studies that included benign and malignant PNETs [2]. Stratification of prognosis in patients with PNET is currently determine by the 2017 World Health Organization grading system using mitotic counts and the Ki-67 levels [3]. The tumor–node–metastasis (TNM) staging system (according to primary tumor size, lymph node [LN] metastasis, and distant metastases) is a prognostic prediction system for all kinds of malignancies, described in the eighth edition of the American Joint Committee on Cancer (AJCC). LN status is particularly crucial in PNET staging systems because LN metastasis would classify a patient as Stage III according to both the AJCC and European Neuroendocrine Tumor (ENET) Society staging systems. Both systems are widely used to predict PNET prognosis; however, they have limitations. The 7th edition of the AJCC TNM system for PNET was identical to that for pancreatic exocrine carcinoma; therefore, it could not fully identify the distinguishing features of PNETs. The ENETS TNM staging system for PNETs is superior to the 7th edition of the AJCC system in clinical practice; however, the prognosis between patients with stage IIIA and IIIB PNETs is not significantly different. Because of these limitations, studies have attempted to optimize the existing PNET TNM staging systems. A modified TNM staging system was developed (mENETS system) that combined the T stage definitions of the ENETS staging system and the AJCC staging definitions. The resultant survival curves provided better prognostic evaluation. The recent 8th edition of the AJCC TNM staging system has a revised T stage definition consistent with those of the ENETS and mENETS systems, it exhibits a good prognostic, predictive value for patients with PNETs. Therefore, the 8th edition AJCC TNM system for PNETs is superior [4,5,6].

Metastatic LN involvement, especially the LN ratio (LNR) derived from the number of metastatic LNs divided by the total number of examined LNs, has recently been recognized as a primary prognostic factor of survival and disease progression in pancreatic malignancies, such as ductal adenocarcinoma as well as intra-ductal papillary mucinous and ampullar carcinomas [7]. The accuracy of the LNR is directly proportional to the number of examined LNs, with a minimum of 15–16 LNs examined according to current guidelines [8,9]. However, even in experienced centers, the number of LNs is variable and often below this threshold, leading to possible disease under-staging. A recent study recommended examining at least 28 LNs [10]. Few series have evaluated the prognostic capability of metastatic LN involvement in the setting of PNETs and how it compares to other systems. The contribution of LN metastasis to the prognosis of this disease is controversial. Some studies suggest that it is independently associated with worse disease-free survival (DFS) [11,12,13] and overall survival (OS) [14,15], whereas others did not find any association with OS [2,16].

Boninsegna and colleagues examined a group of 58 patients with an LNR >0.2 who showed higher recurrence rates [17]. Similar results were published by Hashim and colleagues, who reported that LN involvement in PNETs patients is a predictor of a poor prognosis with lower DFS [11]. Those authors also showed that there is an indication for local lymphadenectomy during surgery in PNETs. Two recent retrospective studies also support this approach. The first study used the Surveillance, Epidemiology, and End Results (SEERS) database, which consists of 896 patients who underwent pancreatectomy with lymphadenectomy (between the years 2004–2011) and found that LNRs ≥ 0.5 were independently associated with worse DFS [18]. In the second study, which also used the SEERs database (years 2004–2018), an LNR >0.16 was found as an independent negative prognostic factor for PNETs for both OS and cancer-specific–survival [19]. However, the benefit of LNR, as well as its exact cutoff, are still unknown.

Nevertheless, pre-operative radiologic and histologic investigations are recommended due to the high specificity of a positive imaging finding and its association with the number of positive LN. Further pre-operative risk stratification is essential because it guides the surgical approach and nonsurgical treatment options as well as follow-up protocol [20,21,22].

The gold standard for managing PNETs is surgery in the case of functioning tumors and non–functioning PNETs ≥ 2 cm. However, unlike other cancerous diseases, surgical intervention may also be considered in cases of metastatic disease. In the case of small non-secreting tumors ≤ 2 cm that harbor excellent prognosis, it raises the question if surgery is always the right approach. The benefits of surgery must be carefully considered against the potential harm it may cause.

In advanced G1/G2 PNETs therapy with somatostatin analogs, mTOR inhibitor (everolimus), VEGF inhibitor (sunitinib), and 177Lu-DOTATATE radionuclide therapy (PRRT) have emerged an evidence-based treatment. Multidisciplinary tumor boards should decide on the treatment plan based on the individual characteristics of the patient status and tumor grade and stage [23,24]. This study aims to assess the LNR role as a prognostic factor and determine the most accurate LNR value, which will predict survival and recurrence in patients with PNETs grades 1–3. This study is based on data from a single tertiary center where a multidisciplinary team treats patients. The treatment approach is uniform and adjusted to the patient’s disease grade, stage, and prognostic risk factors.

## 2. Materials and Methods

### 2.1. Study Population

This retrospective study included patients who underwent surgery for primary PNETs between 1997 and 2017 at the Tel Aviv Medical Center. Those who were lost to follow-up post-surgery were excluded.

#### Treatment Protocol

All patients had a baseline evaluation that included clinical assessment, biochemical blood tests, neuroendocrine markers and imaging studies, computed tomography [CT] scan, and/or magnetic resonance imaging (MRI), Ga-68 somatostatin PET scan, and an endoscopic ultrasound including biopsy. The treatment regime was decided by a multidisciplinary team that consisted of an endocrinologist, endocrine surgeon, and oncologist.

Post-operative follow-up included clinical evaluation imaging and neuroendocrine tumor markers. In the case of residual metastatic recurrent or progressive disease, a further treatment plan was decided by the same multidisciplinary team. The first line of intervention after surgery was somatostatin analogs, the second line consisted of treatment with mTOR inhibitor everolimus or VEGF inhibitor sunitinib. The second or third line was 177Lu-DOTATATE radionuclide therapy (PRRT). The fourth line in progressive disease or rapid progressive or de-differentiated disease was chemotherapy. In suitable cases, local intervention such as radio-ablation chemo-embolization was done.

The demographic and clinical data of the suitable patients were reviewed, including age, sex, tumor characteristics (tumor size, location coded as either head, body, or tail), tumor grade, Ki-67% value, perineural status, lymphovascular invasion, surgical margin, and LN status. The 2017 WHO grading system for gastroenteropancreatic NETs based on mitotic count and Ki-67 proliferation index used. According to that system, Grade 1 corresponds to a Ki-67 value between 0–3%, Grade 2 matches a Ki-67 value between 3–20%, and Grade 3 matches a Ki-67 value above 20% [3]. Both AJCC and ENET Society staging systems were used [4,5]. The study was approved by the Tel–Aviv Medical Center institutional review board 307-14-TLV which waived informed consent.

### 2.2. Statistical Methods

Categorical variables were expressed as frequencies and percentages. Continuous variables were presented as the mean with standard deviation.

An optimal LNR cutoff for events of survival and progression-free survival was found using ROC (receiver operating characteristic )analysis consisting of Area Under the Curve (AUC) computation, followed by maximization of Youden’s index.

Survival analysis was performed for events of overall survival and disease-free survival. Kaplan-Meier curves and Cox Proportional Hazards Model for the above events were used with the optimal LNR cutoff found in the ROC analysis and Ki67 value as studied variables Univariate and multivariate analyses were performed, correcting for known confounders and prognostic factors.

*p* values lower than 0.05 were considered statistically significant. All statistical analyses were performed using SPSS version 25.0 (IBM, Chicago, IL, USA).

## 3. Results

A total of 95 patients with primary PNET who underwent surgical resection between 1997–2017 were identified; 17 of them were excluded due to adequate data or follow-ups. This left 78 patients for inclusion in the final analysis. The study patients’ demographics and clinical characteristics are summarized in Table 1.

The mean ± standard deviation age of the entire cohort was 57.4 ± 13.4 years, the mean follow-up was 4.2 ± 3.4 years; 50 were males (64.1%) and 28 were females (35.9%). The mean ± standard deviation tumor size was 2.5 ± 2.75 cm. Tumor location was 36% at pancreatic head, 23% at body, and 39.7% at tail. At presentation, 7% of the study cohort patients had metastatic disease (Table 1). Forty-four patients (56%) had Grade 1 PNETs, 20 had Grade 2 (26.9%), and 7 had Grade 3 (9%). There was no information on either the exact figure or range of the 67KI value in the pathological reports of 7 patients (9%) (Table 2).

At diagnosis, five patients (7.6%) had distant metastasis in the liver (Stage IV), 23 (29.5%) had LN involvement (N1), 14 (17.4%) had vascular invasion, and ten (12.8%) had a perineural invasion.

### 3.1. Tumor Stage according to AJCC/ENET

Twenty-six (33.3%) of the operated patients had Whipple procedures, 37 (47.4%) had a distal pancreatectomy, 5.1% (*n* = 4) had a complete pancreatic resection, and 11 (14.1%) had tumor excision. The aim of the operation was curative for all patients (Table 1). During follow-up, 18 patients died, and 17 patients experienced recurrence, mostly liver metastasis (*n* = 14, 18%) (Table 3).

The OS and the DFS were calculated relative to the date of surgery. The OS and DFS at 2 years following the surgery were 85.9% and 79.5%, respectively. The OS and DFS at 5 years following the surgery were 82.1% and 71.8%, respectively (Figure 1).

Tumor grade significantly affected the DFS and the OS at 2 and 5 years, and the effect was preserved after adjustment for the selected variables (vascular invasion, smoking diabetes, age, sex, metastatic disease at diagnosis, and adjuvant therapy) (Figure 2). No cutoff for positive LNs was found to have a significant effect on either DFS or OS.

### 3.2. Finding the LNR Threshold

Receiver operating characteristic curves (ROC) were constructed for all four survival measures, and the value that gave an optimal combination of sensitivity and specificity was found to be 0.1603 (Table 4).

The optimal LNR values in our study correlated with the outcome were 2-year OS (*p* = 0.002 HR = 13.4), 5-year OS (*p* = 0.004 HR = 9), and 5-year DFS (*p* = 0.001 HR = 10.6). However, the multivariate analysis failed to show that the LNR was an independent prognostic factor in PNETs (Figure 3a–c).

Further evaluation of the optimal LNR for Grade 1–2 patients showed an optimal LNR value of 0.6–0.7 but with no correlation to OS or DFS. The association between the LNR and OS and DFS for the Grade 3 patients could not be evaluated due to the small sample size.

## 4. Discussion

The overall prognosis of patients with a nonfunctional pancreatic neuroendocrine tumor (NF-pNET) is usually favorable, but it is imperative to detect NF-pNET patients who bear a high risk of recurrence or death. There are still no available biochemical or genetic profiles to provide tools for determining prognosis, the protocol for adjuvant therapy, or the frequency of follow-up. The AJCC and the European Neuroendocrine Tumor Society guidelines for staging and grading TNM are currently used as prognostic indicators of PNETs [4,5], but the contribution of a finding of LN metastasis to the estimation of prognosis is controversial. Some studies suggest that LN metastasis is independently associated with a worse DFS and OS in patients with PNET, whereas others did not find any comparable association with OS. These conflicting results may be due to the variable surgical approaches and the need for lymphadenectomy in cases of pancreatic neuroendocrine tumors [11,12,13,14,15,16].

Studies on pancreatic adenocarcinoma and PNETs have suggested that the LNR may be a more accurate determinant of patient prognosis. Several reports have demonstrated that an LNR >0.07–0.2 was a prognostic factor for DFS, while an LNR of 0.07–0.5 was a prognostic factor for OS [17,18] in patients with PNETs. In the current study, an LNR >0.1603 emerged as being the optimal value for evaluating recurrence and survival risks on univariate analysis, thus critical for those mandating a more aggressive intervention. However, the multivariate analysis was unable to show that an LNR >0.16 was an independent prognostic factor of OS or DFS. The further evaluation also could not demonstrate any association between the LNR and OS and DFS for those with PNETs Grades 1 and 2 (the group with Grade 3 was too small for assessment). We also found that the number of examined LNs failed to demonstrate any survival benefit in the entire cohort. The conflicting results of our study and other studies can be explained by the different studies’ designs and the confounder of tumor characteristics as grade stage tumor size were different in each study. These findings taken together indicate that the LNR should be considered as an additional risk factor during the process of tumor risk assessment, and it does not suffice as an independent risk factor for prognosis in the setting of PNETs, especially not for patients with lower tumor grade Ki-67% or tumor size. The optimal LNR that can be used as a prognostic tool remains to be determined by larger and prospective studies that include data on grade, KI67% status, stage, pathologic information about lymph vascular and perineural invasion, necrosis, tumor size, and uniform approach in terms of assessment, surgical treatment, follow-up, and pharmacological intervention that may determine LNR ratio that is adjusted to all the mentioned confounders.

The current study has several limitations, including the retrospective design and exclusion of cases due to the unavailability of data. Changes in neuroendocrine tumor management protocol over the years as previously somatostatin analogs were used only to control the symptoms associated with carcinoid syndrome. In the last decade, two placebo-controlled trials have addressed the PFS and OS benefits of long-acting somatostatin analogs (PROMID and CLARINET trials), thus treatment with somatostatin analogs was added as an anti-proliferative treatment. Additional treatment options were added in the case of progressive disease, peptide receptor radiotherapy (PRRT), or Sunitinib agent that targets the vascular endothelial growth factor (VEGF) and everolimus, an inhibitor of rapamycin (mTOR) [24,25,26,27,28]. Additionally, the study sample size is small (78 patients), as is the sample of Grade III PNETs patients. The strengths of this study are that it was carried out in a large tertiary center in which patients with neuroendocrine tumors are treated by a multidisciplinary team that provides a uniform standard of care as opposed to large cancer registries, which contain patients from different institutions and time periods and are subject to different treatment protocols. Also in this study, there are data on confounders that according to literature each one of them may affect prognosis.

In summary, LNR failed to be an independent prognostic factor in patients with NF-PNETs, but it can be considered as an additional prognostic factor that should be considered for choosing appropriate adjuvant therapeutic strategies and surveillance protocols for these patients.

## Figures and Tables

**Figure 1 biomedicines-11-00407-f001:**
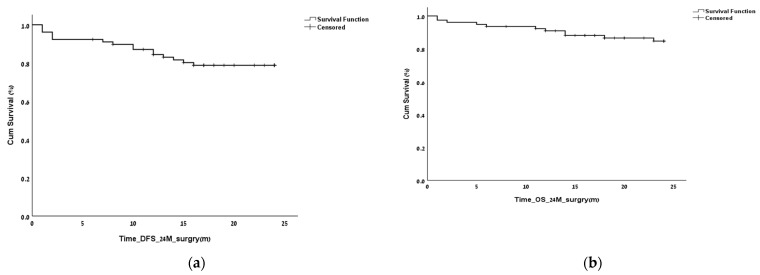
Overall survival (OS) and disease-free survival (DFS) of the entire cohort at 2 and 5 years. OS and DFS at 2 years postoperatively (**a**,**b**) and after 5 years postoperatively (**c**,**d**).

**Figure 2 biomedicines-11-00407-f002:**
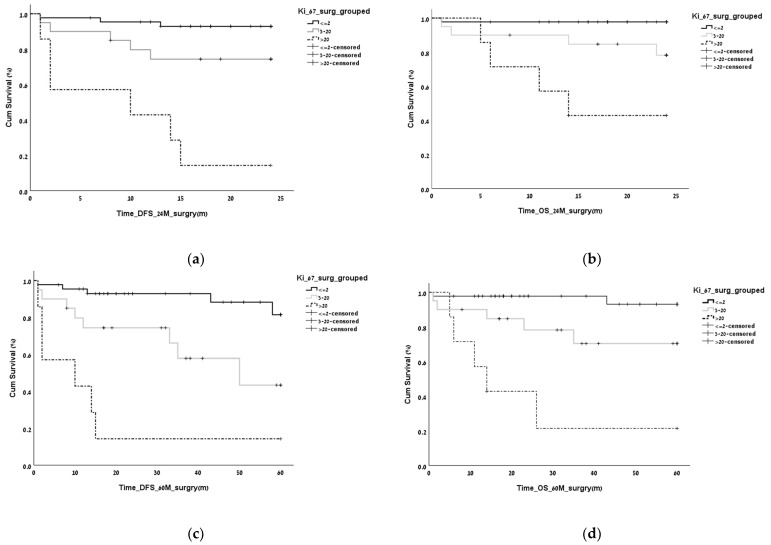
(**a**,**b**) Effect of tumor grade on (**a**). disease-free survival (DFS) and overall survival (OS) at 2 years (*p* < 0.001). The effect was preserved after adjustment for the selected variables (vascular invasion, smoking, diabetes, age, sex, metastatic disease at diagnosis, and adjuvant therapy) (**c**,**d**). Effect of tumor grade on disease-free survival (DFS) and overall survival (OS) at 5 years (*p* < 0.001). The effect was preserved after adjustment for the selected variables (vascular invasion, smoking, diabetes, age, sex, metastatic disease at diagnosis, and adjuvant therapy).

**Figure 3 biomedicines-11-00407-f003:**
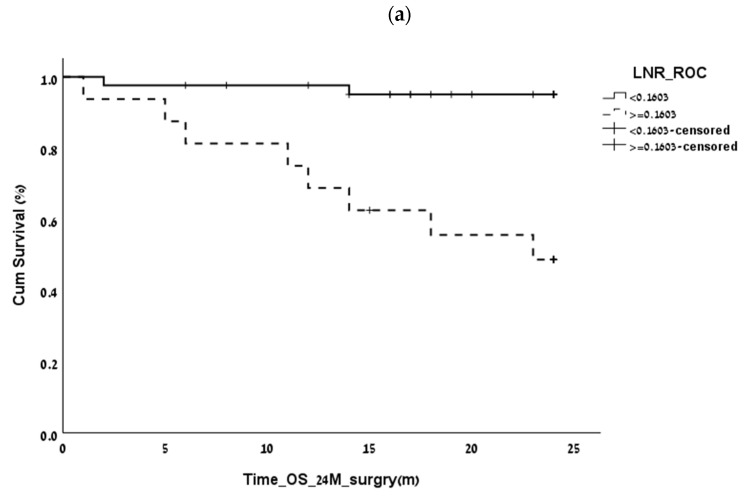
Correlation of lymph node ratio (LNR) with the outcome at 2 and 5 years postoperatively. (**a**). LNR: 2-year overall survival (OS) (*p* = 0.0016). (**b**). LNR: 5-year disease-free survival (DFS) (*p* = 0.001). (**c**). LNR:5-year overall survival (*p* = 0.004).

**Table 1 biomedicines-11-00407-t001:** Demographics of the study cohort.

Variable	
Age, years, mean ± standard deviation Follow-up, mean ± standard deviation	58 ± 13.4 (range 20–82) 4.2 ± 3.4 years
Sex, *n* Male Female	50 (64.1%) 28 (35.9%)
Type of surgery	Whipple procedure (*n* = 26, 33%) Distal pancreatectomy (*n* = 37, 47.4%) Complete pancreatic resection (*n* = 4, 5.1%) Tumor excision (*n* = 11, 14.1%)
Tumor size, cm median ± standard deviation	2.5 ± 2.75
Tumor location Head (%) Body (%) Tail (%) Unknown(%)	36 23 39.7 1.3
Vascular invasion (%) Perineural invasion (%)	17.4 13
Smokers (%) Diabetes mellitus (%)	19.2 37.8
Metastatic disease at presentation (%)	7

**Table 2 biomedicines-11-00407-t002:** Tumor stage and grade distribution.

	American Joint Committee on Cancer	European Neuroendocrine Tumor Society
Stage	I: 35.9% (*n* = 28) II: 24.4% (*n* = 19) III: 25.6% (*n* = 20) IV: 9% (*n* = 7)	I: 19.2% (*n* = 15) II: 20.5% (*n* = 16) III: 43.6% (*n* = 34) IV: 11.5% (*n* = 9)
Tumor grade		1: 56% (*n* = 44) 2: 26.9% (*n* = 20) 3: 9% (*n* = 7) No data: 9% (*n* = 7)

**Table 3 biomedicines-11-00407-t003:** Tumor recurrence in the entire study cohort.

Recurrence Site	
Liver, *n*	14 (18%)
Lung, *n*	2 (2.6%)
Peritoneum, *n*	3 (8%)
Lymph nodes, *n*	2 (2.6%)

Three patients had multiple site involvement.

**Table 4 biomedicines-11-00407-t004:** Lymph node ratio sensitivity and specificity.

Clinical Measure	Optimal Lymph Node Ratio	Sensitivity	Specificity	*p*
2-year overall survival	0.16	0.8	0.83	0.002
2-year disease-free survival	0.16	0.62	0.82	0.023
5-year overall survival	0.16	0.69	0.84	0.04
5-year disease-free survival	0.15	0.6	0.85	0.05

## Data Availability

The datasets generated and/or analyzed during the current study are not publicly available due to patient’s privacy. Personal patient information was anonymized and stored under a password-protected computer. The computer is located at a locked office of the investigator. The data that support the findings of this study are available from Ester Osher, but restrictions apply to the availability of these data, which were used under license for the current study, and so are not publicly available. Data are however available from the authors upon reasonable request and with permission of Tel Aviv Sourasky Medical Center Helsinki Committee.

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
