# Peer review of "A Lymph Node Ratio Model for Prognosis of Patients with Pancreatic Neuroendocrine Tumors"

_biomedicines, 2023, doi:10.3390/biomedicines11020407_

Round 1
Reviewer 1 Report
This study assessed the prognostic value of lymph node ratio on recurrence rates and survival in patients with pancreatic neuroendocrine tumors undergoing surgery. The authors reported LNR was correlated with the 2.5- and 5-year disease-free survival (DFS) rates and the 2- and 5-year overall survival (OS) rates in PNETs. They indicated that LNR fail to be an independent prognostic factor in PNETs. Some concerns need to be explained before considering the publication of this manuscript.
1. Since very similar studies have been reported, the author should stress the novelty and significance of this study.
2. Statistical methods should be described in detail.
3. The approval/document number (title) for IRB is recommended to provide.
4. Nodal invasion is a powerful prognostic indicator in various cancers. However, marked heterogeneity exists within late stage III patients. Therefore, the authors should provide more discussion about the theme.
5. Overall, the research context, methodology and discussion are woefully inadequate.
6. Small editorial corrections such as periods, commas, spaces, superscripts and subscripts are required.
Author Response
The response to the reviewer attached in the attached file

Reviewer 2 Report
This retrospective study investigates an important point: the impact of limphnode ratio on disease recurrence in patients with PanNETs who undergo surgical resection.
Here are listed my suggestions:
- Table 1 should be implemented with other baseline features (e.g., size and site of the lesion)
- There are no data about the correlation of lesion size with recurrence
- Also, other known prognostic factors should be considered (e.g., vascular invasion, perineural invasion)
- A multivariate analysis should be performed to assess the independency of these factors.
- The importance of preoperative evaluation and LN ration should be mentioned. Doing so, cite: PMID: 35863518; PMID: 35930017; PMID: 33508821; PMID: 33177357
Author Response
The response to the reviewer comments attached at the file

Reviewer 3 Report
It is no news that metastasis in lymph nodes is no prognostic algorithm in evolution of PNET patients. I think that is important to see that is there is a connection between the number of positive lymph nodes and stage and survival.
Reviewer 4 Report
The introduction is comprehensive and exposes current issues related to pancreatic neuroendocrine tumors and the possibility of using LNR as a prognostic factor for OS and DFS.
The methodology of the study is well-defined, the patients included in the study come from a tertiary referral center with a focus on neuroendocrine pathology, which adds value. Results are presented accurately using advanced statistical methods.
However, I cannot help but notice that this is a retrospective study extending back to 1997, and references are made to the TNM AJCC 8th edition classification, which is currently used. As a suggestion, that would provide an interesting perspective- could the authors of the study comment a little on the T changes in the TNM classification and how these changes might influence the study's conclusions.
Also under limitations, you could add the changes in neuroendocrine tumor management protocol over the years, and maybe you might elaborate on it. I would also suggest adding some references from the last decade.
Otherwise, it is a very well written article from an experienced center in the field that I hope will present us with further results in the near future.
Round 2
Reviewer 1 Report
All concerns have been satisfied and the manuscript can be accepted in the present form after checking the language.
Reviewer 2 Report
I have no further comments